# Microstructure and High-Temperature Properties of Cr₃C₂-25NiCr Nanoceramic Coatings Prepared by HVAF

Zhiqiang Zhou [1], Dajun Duan [1,*], Shulan Li [1], Deen Sun [2], Jiahui Yong [1], Yongbing Jiang [1], Wu He [1] and Jian Xu [1]

1   Chongqing Chuanyi Control Valve Co., Ltd., Chongqing 400707, China; zhouzzq6@gmail.com (Z.Z.)
2   School of Materials and Energy, Southwest University, Chongqing 400715, China; deen_sun@cqu.edu.cn
*   Correspondence: ddj@siccv.com

**Abstract:** The study examines the microstructure and high-temperature properties of Cr₃C₂-25NiCr nanoceramic coatings on 316H high-temperature-resistant stainless steel that were prepared by high-velocity air–fuel spraying (HVAF) technology. The micromorphology, phase composition, fracture toughness, high-temperature hardness, high-temperature friction, and wear properties of the coating were studied by scanning electron microscopy (SEM), X-ray diffraction (XRD), high-temperature Vickers hardness tester, high-temperature friction and wear tester, and surface profiler. The results show that the Cr₃C₂-25NiCr coating prepared by HVAF presents a typical thermal spraying coating structure, with a dense structure and a porosity of only 0.34%. The coating consists of a Cr₃C₂ hard phase, a NiCr bonding phase, and a small amount of Cr₇C₃ phase; The average microhardness of the coating at room temperature is 998.8 $HV_{0.3}$, which is about five times higher than that of 316H substrate. The Weibull distribution of the coating is unimodal, showing stable mechanical properties. The average microhardness values of the coating at 450 °C, 550 °C, 650 °C, and 750 °C are 840 $HV_{0.3}$, 811 $HV_{0.3}$, 729 $HV_{0.3}$, and 696 $HV_{0.3}$ respectively. The average friction coefficient of the Cr₃C₂-25NiCr coating initially decreases and then increases with temperature. During high-temperature friction and wear, a dark gray oxide film forms on the coating surface. The formation speed of the oxide film accelerates with increasing temperature, shortening the running-in period of the coating. The oxide film acts as a lubricant, reducing the friction coefficient of the coating. The Cr₃C₂-25NiCr coating exhibits exceptional high-temperature friction and wear resistance, primarily through oxidative wear. The Cr₃C₂-25NiCr coating exhibits outstanding high-temperature friction and wear resistance, with oxidative wear being the primary wear mechanism at elevated temperatures.

**Keywords:** HVAF; Cr₃C₂-25NiCr coating; micromorphology; mechanical properties; high-temperature friction and wear properties

## 1. Introduction

316H stainless steel is frequently employed as a high-temperature-resistant material for valve stem components in high-temperature, high-pressure steam conditions, with a maximum operating temperature reaching 816 °C. Nevertheless, at elevated temperatures, the hardness of 316H stainless steel is relatively low, rendering it susceptible to erosion caused by gas or particulate-laden gas-solid two-phase media. Consequently, there is a need to elevate the surface hardness of 316H stainless steel through surface-hardening techniques.

Cr₃C₂-25NiCr currently stands as one of the most extensively employed nanoceramic composite materials, among which the NiCr alloy component of this composite boasts commendable resistance to heat-induced corrosion and exceptional capabilities in countering high-temperature oxidation. The elevated hardness attributed to Cr₃C₂ further enhances its exceptional resistance to high-temperature friction and wear [1]. Consequently, the Cr₃C₂-25NiCr coating exhibits commendable performance in withstanding elevated temperatures, displaying resilience against high-temperature erosion, oxidation, and wear

at 900 °C. This prowess has propelled its successful integration across industries encompassing petroleum refining, thermal power generation, aerospace, metallurgical machinery, and other fields [2,3]. Notably, thermal spray technology emerges as a pivotal technique in the fabrication of $Cr_3C_2$-25NiCr coatings [4].

The utilization of high-velocity oxygen fuel spraying (HVOF) technology is prevalent in the preparation of $Cr_3C_2$-25NiCr coatings. However, the HVOF technology uses oxygen as the combustion-assisting gas, and the metal-powder particles are in a rich oxygen atmosphere during the spraying process, which is prone to thermal decomposition of powder oxidation or carbides [5]. High-velocity air–fuel spraying (HVAF) technology is a new technology developed in recent years. HVAF uses compressed air instead of expensive oxygen as the combustion-assisting gas and adopts a gas-cooling method. HVAF has a higher spraying flame velocity and lower flame temperature than HVOF technology, which helps to form metal coatings with high density, low oxide content, and high bonding strength [6].

Mathiyalagan et al. [7] utilized the HVAF technique to fabricate Ni-P coatings containing c-BN on the surface of 350LA, resulting in a hardness enhancement of 47% and a reduction of the wear rate by two orders of magnitude. The wear resistance was significantly improved. The dimensions of the combustion chamber also exert an influence on the wear resistance of the coating. The research findings indicate that a relatively larger combustion chamber can notably diminish the quantity of nondeformed particles within the coating, thereby leading to lower porosity and relatively higher hardness within the coating, thus achieving the objective of enhancing the coating's wear resistance [8]. Alroy [9] investigated the influence of process parameters on the corrosion performance of HVAF-prepared $Cr_3C_2$-25NiCr coatings. It was observed that by using fine-grained powders and a medium-length nozzle during the coating preparation, the coating exhibited reduced porosity and higher density, leading to a corrosion rate decrease of 40%–45% compared to the substrate. Furthermore, the spray flame velocity and flame temperature of HVAF were reported to be 700–1200 m/s and 1800 °C, respectively [10,11]. The semimolten sprayed powders have a very short flight time in the air, which can effectively suppress oxidation, decomposition, and decarburization of the powder so that the majority of hard phases can be retained. Therefore, HVAF technology has received a lot of attention [12,13].

In electric power, metallurgy, and other industries, various ball valves are installed on pipelines as switching control equipment for fluid media. The temperature of high-temperature heat transfer oil, high-temperature flue gas, high-temperature steam, and other media in the pipeline is often as high as about 600 °C, and its key parts are easily damaged and fail, seriously affecting the normal operation of the ball valve. Therefore, the surface strengthening of the critical components is very important. Usually, researchers will spray a hardened layer on the ball core, valve seat, and other seals of the ball valve to ensure the high-temperature oxidation resistance and wear resistance of the valve under high-temperature conditions. However, at present, there is limited research on the high-temperature performance of $Cr_3C_2$-25NiCr nanoceramic coatings prepared using HVAF technology. Currently, there is limited research on the high-temperature performance of $Cr_3C_2$-25NiCr nanoceramic coatings prepared using HVAF technology. In order to enhance the high-temperature surface-wear resistance of AISI 316H stainless steel, based on HVAF technology and optimized process parameters, $Cr_3C_2$-25NiCr nanoceramic coatings were prepared on the surface of a 316H stainless steel substrate. Through a series of experiments, the microstructural morphology, phase composition, high-temperature hardness, and high-temperature friction-wear performance of the coating were investigated. Additionally, the high-temperature friction-wear mechanism was explored, thereby establishing the feasibility of applying the $Cr_3C_2$-25NiCr nanoceramic coating prepared by HVAF technology under high-temperature operating conditions.

## 2. Materials and Methods

### 2.1. Coating Preparation

The substrate material for the coating preparation is 316H stainless steel; its chemical composition is shown in Table 1. The chemical composition of the substrate was determined using ICAP 7000 inductively coupled plasma optical emission spectrometry (ICP-OES, Thermo Fisher Scientific, Waltham, MA, USA) and a CS-988A carbon sulfur analyzer (CSA, Wuxi Tianmu Instrument Technology Co., Ltd., Wuxi, China). The specimen size was 30 mm × 15 mm × 8 mm. To remove any impurities, such as oxide or oil residues that may remain on the surface of the sample, the 316H substrate was repeatedly cleaned using acetone and absolute ethanol and dried with compressed air. After cleaning, the substrate was sandblasted to ensure optimal surface roughness, thereby enhancing the bond strength between the coating and the substrate.

**Table 1.** Chemical composition of 316H stainless steel (wt.%).

| C | Mn | Si | S | P | Ni | Cr | Mo | Fe |
|---|----|----|----|---|----|----|----|----|
| 0.07 | 1.83 | 0.86 | 0.02 | 0.03 | 13.25 | 17.15 | 2.31 | Bal. |

The spraying powder used for the experiments is a $Cr_3C_2$-25NiCr powder produced by Luoyang Jinglu Hard Alloy Tool Co., Ltd. (Luoyang, China). The powder is manufactured using a method involving Ni and Cr encapsulation of $Cr_3C_2$, and its composition is detailed in Table 2. Powder particle-size distribution is measured using a laser particle-size analyzer (Microtrac S3500, Boca Raton, FL, USA).

**Table 2.** Chemical composition of $Cr_3C_2$-25NiCr powder (wt.%).

| C | Ni | Fe | Cr |
|---|----|----|----|
| 10.66 | 20.35 | 0.41 | Bal. |

The coating preparation is conducted using the M2 HVAF (UNIQUECOAT, Oilville, VA, USA) supersonic flame-spraying system. In this experiment, the application employs optimized spray-process parameters for batch production (as presented in Table 3) targeting the fabrication of $Cr_3C_2$-25NiCr nanoceramic coatings on 316H stainless steel.

**Table 3.** Process parameters of HVOF and HVAF.

| Air pressure (MPa) | Propane pressure (MPa) | Nitrogen flow (L/min) | Airflow ($m^3$/min) | Spraying distance (mm) | Powder speed (mm/s) | Powder feeding (g/min) |
|---|---|---|---|---|---|---|
| 0.54 | 0.49 | 60 | 20 | 230 | 800 | 110 |

Coating cross-section specimens are prepared using a wire-cutting process. Following thermal embedding, coarse grinding, fine grinding, and polishing, the surface roughness of the coating is achieved at Ra < 0.2 μm.

### 2.2. Performance Characterization

The microstructure of $Cr_3C_2$-25NiCr powder and coatings was observed using the Navo Nano SEM 450 field emission scanning electron microscope (SEM, FEI, Hillsboro, OR, USA). Additionally, the chemical composition of the coating was analyzed by Quantax-200 EDS (Bruker, Billerica, MA, USA) spectrometry.

The coating cross-section microstructure is examined using the Axio Observer 3 m (Carl Zeiss AG, Oberkochen, Germany) research-grade metallographic microscope. The average porosity of 5 different fields is calculated using the Pro Imaging metallographic intelligent analysis system.

The phase composition of $Cr_3C_2$-25NiCr powder and the coatings is analyzed using the Rigaku D/MAX 2500 PC X-ray diffractometer (XRD, Rigaku, Tokyo, Japan). The samples are subjected to phase testing with a tube voltage of 30 kV, tube current of 20 mA, scanning angle range of 20°–90°, and continuous scanning rate of 0.03 °/s.

Microhardness of the coatings was measured using the INNOVATEST FALCON 500 (Eindhoven, The Netherlands) Vickers hardness tester with a load of 300 g and a loading time of 15 s. Additionally, 15 hardness values are obtained in the field-of-view area of the coating cross-section, and the Weibull distribution method is employed to explore the hardness distribution characteristics of the coating at room temperature. Equations (1) and (2) were employed to characterize the hardness-distribution characteristics of the coating [14].

In the equation, where $F(x)$ represents the hardness probability distribution function of the coating, $m$ stands for the modulus of the Weibull distribution function, reflecting the degree of discreteness of numerical expressions. The parameter $x$ represents the measured hardness value, while $x_0$ signifies the hardness value obtained after sorting the hardness points in ascending order, accounting for 63.2% of the total hardness points.

$$\ln\{\ln[\frac{1}{1 - F(x)}]\} = m[\ln(x) + \ln(x_0)] \tag{1}$$

$$F(x) = \frac{i - 0.5}{n} \tag{2}$$

In the equation, where $n$ represents the total number of hardness indentations, and $i$ indicates the sequence number of hardness values arranged in ascending order.

The high-temperature microhardness of the coatings is evaluated using the HVZHT-30 high-temperature vickers hardness tester (ZONE-DE, Shandong, China), applying a load of 500 g and a dwell time of 10 s. The holding time at each temperature is 10 min. Three hardness values are recorded at each temperature, and real-time high-speed images of the indentations are captured.

The UMT-3 multifunctional high-temperature friction and wear tester (CETR, San Jose, CA, USA) is utilized to assess the friction-wear performance of the $Cr_3C_2$-25NiCr coating in high-temperature air at temperatures of 450 °C, 550 °C, 650 °C, and 750 °C, respectively. The test is pin–disc contact, and for every single test, a test duration of 30 min and a new $Al_2O_3$ (the Mohs hardness scale is 9) ball with a diameter of 9.5 mm were used, with a load of 5 N and a disc rotation speed of 150 r/min.

The DektakXT (Bruker, Billerica, MA, USA) probe surface profiler measures the cross-sectional profiles of the wear scars.

## 3. Results

### 3.1. Powder Morphology

The microstructure of the agglomerated $Cr_3C_2$-25NiCr spherical nanopowder for supersonic spray prepared through the gas atomization (agglomerating sintering) process is shown in Figure 1a. It is evident that the powder possesses a high degree of sphericity and excellent dispersion, indicating a favorable flowability of the powder.

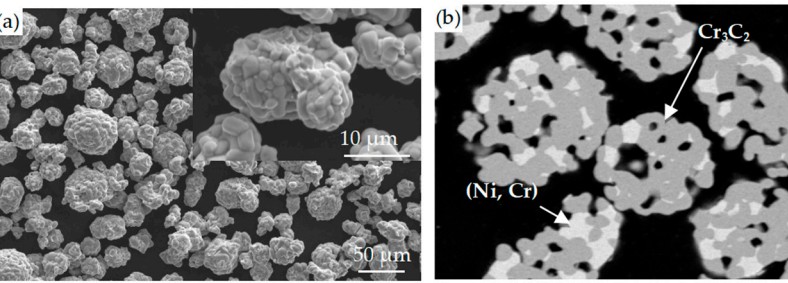

**Figure 1.** BSE image of $Cr_3C_2$-25NiCr powder (**a**) surface morphology; (**b**) cross-sectional morphology.

Figure 1b shows the powder cross-sectional metallurgical structure. The metallurgical structure within the cross-section reveals the $Cr_3C_2$ phase wetted by the NiCr alloy $\gamma$ phase [15]. The gray region corresponds to the $Cr_3C_2$ phase, while the silver-white region represents the NiCr alloy $\gamma$ phase. It can be observed that the $Cr_3C_2$ particles in the powder were sintered, and the NiCr alloy bonding phase uniformly filled the interstices between the hard particles, forming a stable skeleton-network structure.

Figure 2 illustrates the particle-size distribution of the powder, conforming to a standard Gaussian distribution. Among the parameters, dmean = 32 μm, d10 = 18 μm, and d90 = 52 μm.

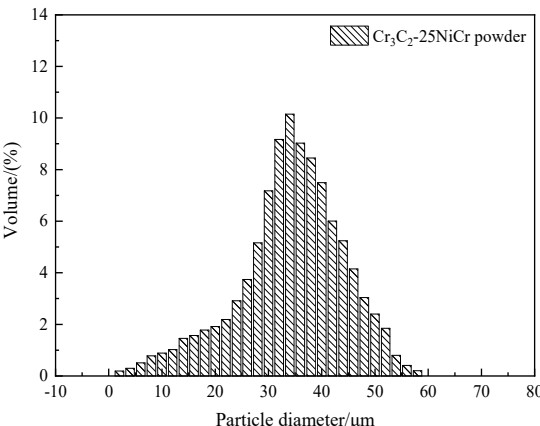

**Figure 2.** Particle-size distribution of $Cr_3C_2$-25NiCr powder.

*3.2. XRD Phase Composition*

Figure 3 exhibits the X-ray diffraction (XRD) spectra of $Cr_3C_2$-25NiCr coatings prepared by HVAF, as well as the initial powder. It is evident that the sprayed coating retains the crystalline phases present in the raw powder. The $Cr_3C_2$-25NiCr powder and coating are primarily composed of the $Cr_3C_2$ hard phase and NiCr bonding phase. Additionally, a small amount of the $Cr_7C_3$ diffraction peak is observed in the coating. This can be attributed to the partial decarburization of $Cr_3C_2$ during the high-temperature flame in the spraying process [16]. Both $Cr_3C_2$ and $Cr_7C_3$ possess high hardness and high melting points, which contribute to the wear resistance and high-temperature hardness of the coating [3].

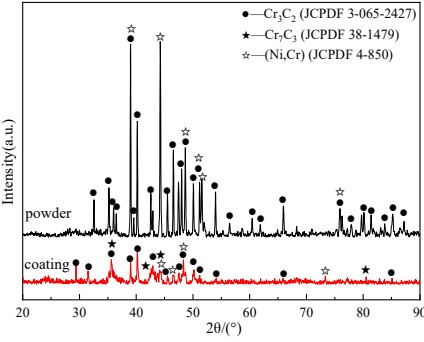

**Figure 3.** XRD pattern of $Cr_3C_2$-25NiCr powder and coating.

Notably, there is a substantial difference in the intensity of diffraction peaks between the powder and the coating. The diffraction peaks of the coating exhibit a significant broadening phenomenon, indicating the generation of an amorphous phase during the transformation from powder to coating. This can be attributed to the rapid cooling of molten or semimolten droplets upon reaching the substrate during the supersonic spray process. This rapid cooling suppresses crystal growth, resulting in a disordered accumulation of solidifications and an amorphous coating appearance [17]. Additionally, research suggests

that the formation of the amorphous phase during HVAF spraying of $Cr_3C_2$-25NiCr coatings is related to the severe plastic deformation occurring upon the high-velocity impact of powder particles on the substrate [18]. Consequently, in the process of preparing ceramic coatings through supersonic flame spraying, the coating consists of both crystalline and amorphous phases, with the presence of the amorphous phase potentially enhancing wear resistance to a certain extent [19].

*3.3. Section Morphology*

Figure 4 shows the SEM cross-sectional morphology of $Cr_3C_2$-25NiCr nanoceramic coatings prepared by HVAF. From Figure 4a, it can be observed that the thickness of the $Cr_3C_2$-25NiCr coating is approximately 260 µm, forming a mechanical connection with the substrate through mechanical interlocking. The coating exhibits density without visible cracks or significant defects. During HVAF spraying, with a flame temperature of 1800 °C, the NiCr alloy, due to its lower melting point, rapidly melts and wets the surrounding ultrafine $Cr_3C_2$ nanoparticles, leading to deformation and melting of the outer layers of the $Cr_3C_2$ particles. These molten particles impact the substrate at high speed, flattening and forming a typical thermal spray layer structure. Simultaneously, a small fraction of $Cr_3C_2$ undergoes decarburization, with carbon atoms diffusing into the molten NiCr phase to form a solid solution [20,21]. The coating microstructure is characterized by a continuous and well-melted NiCr bonded phase uniformly dispersed with carbide hard particles, such as $Cr_7C_3$, among which the light gray phase corresponds to the NiCr bonded phase, while the dark gray phase represents the hard particles.

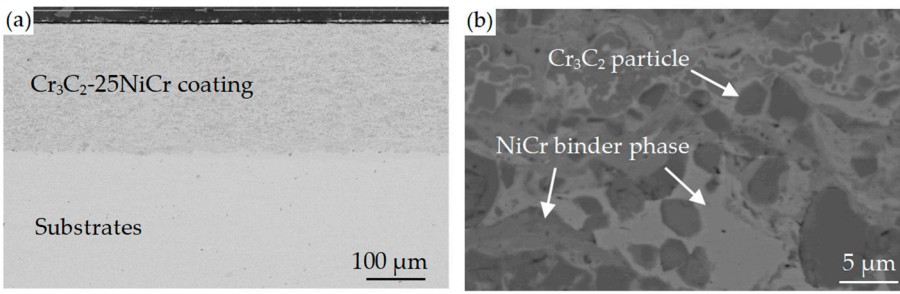

**Figure 4.** BSE cross-section micrographs of $Cr_3C_2$-25NiCr coating: (**a**) lower magnification; (**b**) higher magnification.

Porosity is one of the significant parameters of coating performance, which will significantly affect the microhardness, wear resistance, and corrosion resistance of the coating. The pores of the coating are mainly distributed at the boundaries of hard particles, which is primarily due to insufficient deformation of powder particles and incomplete overlap during deposition. Another portion of the pores is caused by the solidification shrinkage of the fully melted binder phase that cannot be compensated. Employing the "gray level method", the calculated porosity of the coating is 0.34%, indicating a high hardness value [22].

Figure 5 displays the as-sprayed surface morphology of the $Cr_3C_2$-25NiCr coating. It is apparent that the surface of the coating is primarily composed of a multitude of unmelted powder particles along with a small fraction of the completely melted solidification area. This is due to the fact that the HVAF spraying technique employs propane–air as fuel. Compared to HVOF supersonic flame-spraying technology, which uses aviation kerosene as fuel, the heat during the spraying process is low. As a result, some powder particles find it challenging to attain their melting points, and molten or semimolten particles in the flame flow are in a solid–liquid mixed state before impacting the substrate. In the case of HVAF technology, the high powder-injection velocity leads to the flattening of most unmelted or semimelted powder particles. These particles are then layered and bonded

onto the substrate surface through the application of significant impact forces and plastic deformation pressures [23].

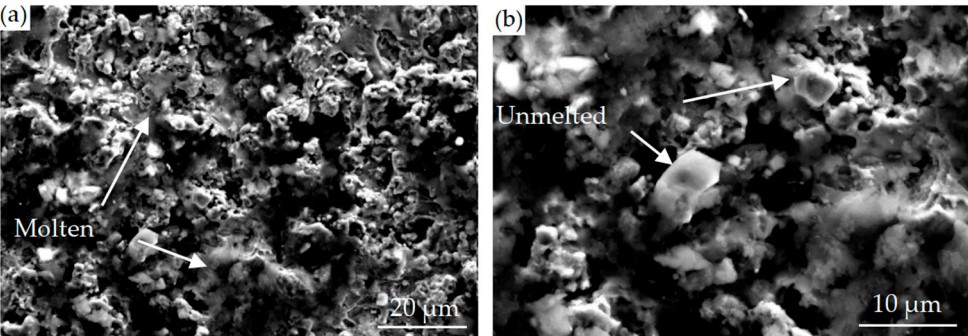

**Figure 5.** As-spraying surface morphology of $Cr_3C_2$-25NiCr coating: (**a**) low magnification; (**b**) high magnification.

### 3.4. Microhardness and Weibull Distribution

Figure 6 depicts the microhardness distribution across the cross-section of the $Cr_3C_2$-25NiCr nanoceramic coating. The average microhardness of the F316H substrate is approximately 210 $HV_{0.3}$, and the microhardness of the substrate near the coating interface tends to increase slightly, reaching around 400 $HV_{0.3}$. This increase can be attributed to the "shot peening effect" induced by the powerful impact of the powder particles on the substrate during a high-speed deposition process. Therefore, the effect of work hardening is produced [24]. The result shows that the closer to the coating interface, the higher the microhardness of the substrate. And at the coating-substrate interface, the microhardness reached 750 $HV_{0.3}$. As can be seen from Figure 6, with the increase in coating thickness, the hardness value of the coating increases slightly and then decreases. It spans from 921 $HV_{0.3}$ to 1090 $HV_{0.3}$, and the average microhardness is about 998 $HV_{0.3}$, which is about five times higher than that of the F316H substrate

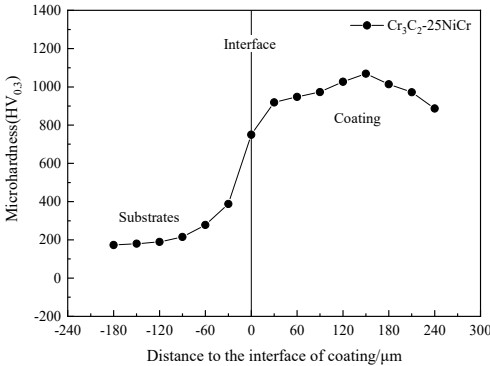

**Figure 6.** Microhardness profile of $Cr_3C_2$-25NiCr coatings.

The Weibull distribution parameters and the Weibull distribution curve of microhardness of the $Cr_3C_2$-25NiCr coating at room temperature are shown in Table 4 and Figure 7, respectively. The distribution of Weibull hardness points of the coating aligns well with the linear fit and demonstrates a unimodal distribution characteristic [25]. The microhardness values of the coating exhibit a narrow distribution range and small hardness extreme values, signifying high uniformity and stable mechanical performance.

**Table 4.** Weibull distribution parameters of microhardness of $Cr_3C_2$-25NiCr coating.

| Hardness(x) (HV$_{0.3}$) | ln(x) | m | n | Average hardness (HV$_{0.3}$) |
|---|---|---|---|---|
| 921~1090 | 6.83~6.99 | 24.45 | 15 | 998 |

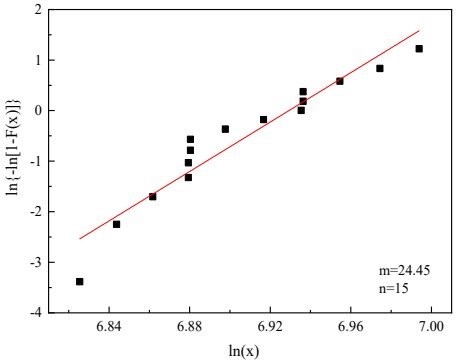

**Figure 7.** Weibull distribution of microhardness for $Cr_3C_2$-25NiCr coating.

*3.5. High-Temperature Hardness*

As indicated in Figure 8, the average microhardness of the $Cr_3C_2$-25NiCr coating at room temperature, 450 °C, 550 °C, 650 °C, and 750 °C are 998 HV$_{0.3}$, 840 HV$_{0.3}$, 811 HV$_{0.3}$, 729 HV$_{0.3}$, and 696 HV$_{0.3}$, respectively. With the increase in temperature, the hardness of the coating decreases. This phenomenon can be attributed to the reduction in both the grain and the grain boundary strength of the material as temperature rises, leading to a high-temperature softening of the coating [26]. Remarkably, even with a temperature increase from 450 °C to 750 °C, the high-temperature hardness of the $Cr_3C_2$-25NiCr coating only experiences a decrease of approximately 140 HV$_{0.3}$, and it still has a high high-temperature hardness.

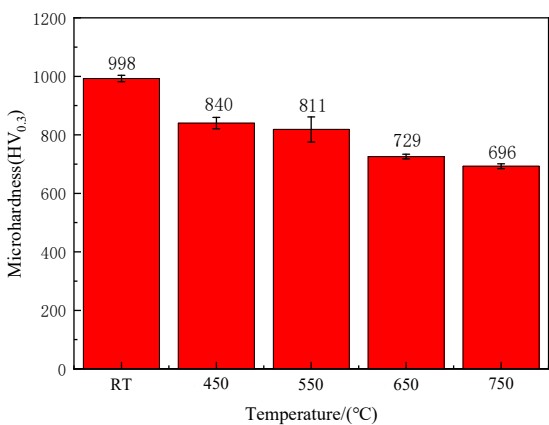

**Figure 8.** Hardness of $Cr_3C_2$-25NiCr coatings at high temperature.

Figure 9 illustrates the real-time microhardness indentation morphology of the $Cr_3C_2$-25NiCr coating at elevated temperatures of 450 °C, 550 °C, 650 °C, and 750 °C. Notably, the indentation morphologies and sizes appear relatively consistent across the temperatures, and no cracks are observed around the diagonals of the indentations. This observation indicates that the $Cr_3C_2$-25NiCr coating maintains a high level of high-temperature hardness and fracture toughness.

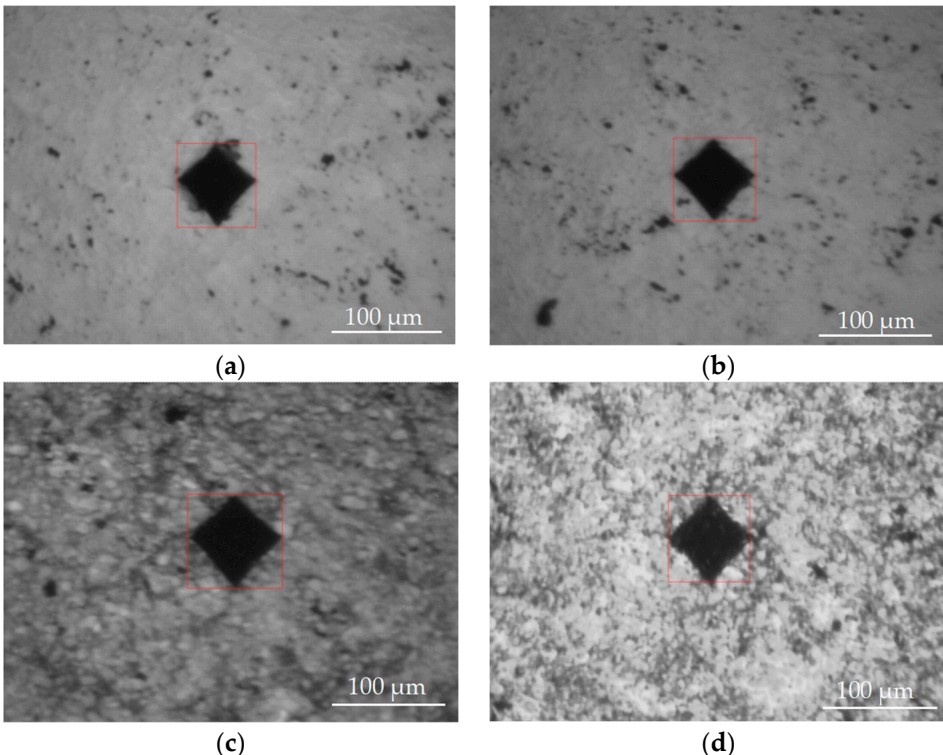

**Figure 9.** Indentation Morphology of $Cr_3C_2$-25NiCr coating at high temperature: (**a**) 450 °C; (**b**) 550 °C; (**c**) 650 °C; (**d**) 750 °C.

With the increase in temperature, the contrast between the light gray $Cr_3C_2$ particles and the surrounding dark gray NiCr bonding phase becomes more distinct, rendering the morphology more discernible under optical microscopy. This phenomenon is attributed to the oxidation of Cr elements on the surface of the coating at high temperatures, resulting in the formation of an oxide film that significantly enhances the contrast between the hard phase and the bonding phase.

### 3.6. High-Temperature Friction and Wear
#### 3.6.1. Coefficient of Friction

Figure 10 illustrates the friction coefficient curves and the average coefficient of friction for the $Cr_3C_2$-25NiCr coating at temperatures of 450 °C, 550 °C, 650 °C, and 750 °C. During the friction test, the friction coefficient of the coating experiences an initial rise followed by a decrease during the running-in period, after which it enters a stable period of small fluctuations. As the temperature increases, the running-in period gradually shortens. At 450 °C, 550 °C, and 650 °C, the running-in periods for the coating are 1056 s, 1010 s, and 987 s, respectively. At 750 °C, the running-in period is only 242 s and swiftly enters a stabilization period. With a rising temperature, the running-in period tends to shorten, and the friction and wear process rapidly enters a stable state.

Notably, at 450 °C, the $Cr_3C_2$-25NiCr coating exhibits the highest coefficient of friction, with an average value of 0.52 ± 0.02. At 550 °C, the lowest coefficient of friction is observed, with an average value of 0.44 ± 0.01. However, at 750 °C, the coefficient of friction increases to 0.48 ± 0.01. Moreover, at this temperature, the friction coefficient was the most stable, fluctuating only within a small range. And with the increase in sliding time, it shows an upward trend.

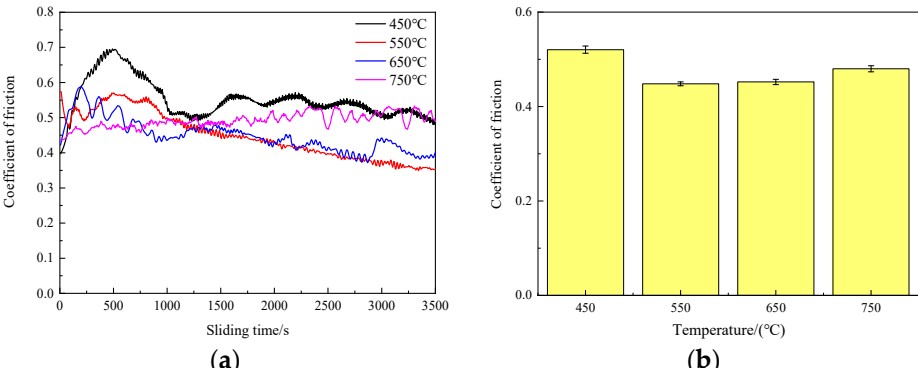

**Figure 10.** Friction coefficient curve and average friction coefficient of $Cr_3C_2$-25NiCr coatings at different temperatures: (**a**) friction coefficient curve; (**b**) mean friction coefficient.

The friction force of the material at high temperatures is mainly the plastic deformation of the surface and the role of the contact point between the friction pair and the surface of the material. With the increase of the test temperature, the plastic deformation force diminishes while the effect of contact points on the surface grows. At high temperatures, adhesive bonding occurs between the friction pair and the material at contact points on the surface. During sliding, the bonded points are sheared, and material transfer takes place at the interface. This alternating process of bonding and shearing causes minor oscillations in the friction curve [27,28]. After entering the stabilization period, the coefficient of friction stabilizes, indicating that the coating has not undergone wear failure. Consequently, the $Cr_3C_2$-25NiCr coating can play a certain protective role on the 316H substrate at a temperature of 450 °C–750 °C.

Figure 11 illustrates the corresponding cross-sectional profile curves and wear rates of the $Cr_3C_2$-25NiCr coating at temperatures of 450 °C, 550 °C, 650 °C, and 750 °C. It can be seen from the figure that, at 450 °C, the $Cr_3C_2$-25NiCr coating has the smallest abrasion area and the lowest wear rate, showing the best wear resistance. With an increasing temperature, the wear rate of the $Cr_3C_2$-25NiCr coating gradually rises, and the corresponding abrasion area gradually increases.

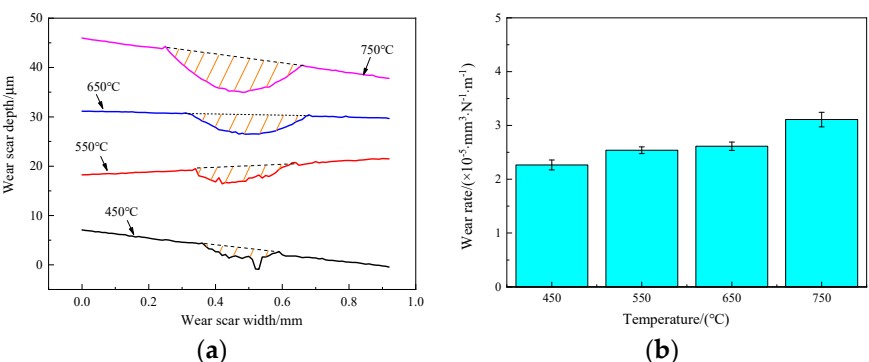

**Figure 11.** The wear scars profile of cross-section and wear rate of $Cr_3C_2$-25NiCr coatings at different temperatures: (**a**) wear-scars profile of cross-section; (**b**) wear rate.

At 450 °C, although the coating exhibits the highest friction coefficient (0.527), its hardness is also the highest (840 $HV_{0.3}$), resulting in a wear rate of $(2.16 \pm 0.03) \times 10^{-5}$ mm$^3$/(N·m). At 550 °C and 650 °C, the wear rates are $(2.52 \pm 0.01) \times 10^{-5}$ mm$^3$/(N·m) and $(2.68 \pm 0.01) \times 10^{-5}$ mm$^3$/(N·m), respectively, and the wear rates are very close, which is related to the minimal variation in its coefficient of friction, suggesting stable mechanical performance of the coating at these temperatures.

However, at 750 °C, the wear rate increases to $(2.97 \pm 0.02) \times 10^{-5}$ mm$^3$/(N·m). This increase is attributed to the higher tendency of softening in the NiCr bonding phase of

the coating at this temperature [29], which reduces the cohesive strength of the coating. As a result, $Cr_3C_2$ (EDS result is shown in Figure 12h) undergoes secondary precipitation, leading to the detachment of hard particles [30]. The coating's shear resistance diminishes, resulting in the highest wear rate observed at this temperature.

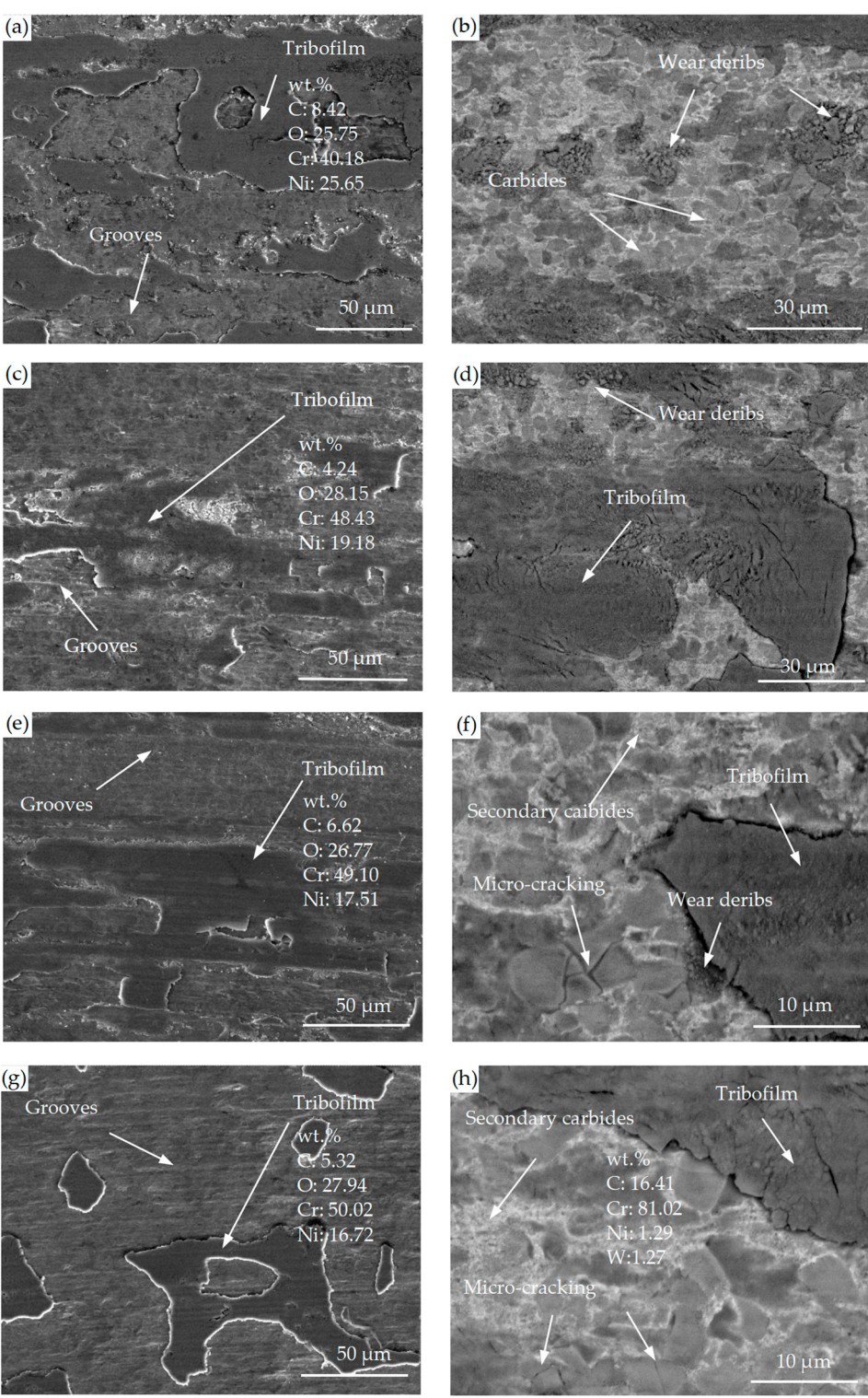

**Figure 12.** SEM micrographs of the wear scars of $Cr_3C_2$-25NiCr coating after wear testing: (**a**,**b**) low- and high-magnification picture at 450 °C; (**c**,**d**) low- and high-magnification picture at 550 °C; (**e**,**f**) low- and high-magnification picture at 650 °C; (**g**,**h**) low- and high-magnification picture at 750 °C.

### 3.6.2. Wear Mechanism

Figure 12 shows the wear-surface morphology of the $Cr_3C_2$-25NiCr coating at different temperatures. The surface exhibits distinctive furrows and extensive areas covered by flaky dark gray oxide film [31]. EDS analysis reveals that these oxide films primarily consist of the elements C, Cr, O, and Ni, of which the oxygen content is 25 wt.%–28 wt.%. This indicates that during high-temperature and oxidative friction and wear, a smooth oxide film (mainly $Cr_2O_3$ and $CrO_3$) is first formed on the surface of the $Cr_3C_2$-25NiCr coating [32].

These oxide-based friction films act as a nonplastic medium, existing between the coating surface and aluminum oxide balls, effectively preventing direct contact and reducing the friction coefficient. Additionally, the oxide-based friction films act as lubricants, further lowering the friction coefficient of the coating [33]. As a result, as shown in Figure 11, the coefficient of friction of the coating decreases when the temperature is 450 °C. With the increasing temperature, the running-in period of the coating shortens gradually. This is due to the accelerated formation rate of the oxide films containing metal Cr on the surface of the coating, and the oxide film plays a lubricating role between the friction pairs, leading to a rapid transition to the stable wear period.

Furthermore, it can be seen in Figure 12b,d that numerous fine debris particles are observed on the coating surface, which play a role in load distribution and also offer a protective effect; it will also have a protective effect and reduce the wear of the coating.

The friction and wear mechanism of the $Cr_3C_2$-25NiCr coating at high temperatures are shown in Figure 13. During high-temperature friction and wear, the formation and breakage of oxide films are in a dynamic equilibrium. In the process, the surface of the preferentially formed sheet of oxide film produces microcracks (as seen in Figure 12 f,h), and gradually loosens and detaches. Then, new fine debris is formed, and the newly exposed alloy area continues to be oxidized after the oxide film is peeled. The peeled debris recombines with the newly formed oxide film, creating a dynamic process of alternating oxidation and detachment. Therefore, at high temperatures, the primary wear mechanism of the $Cr_3C_2$-25NiCr coating is oxidative wear.

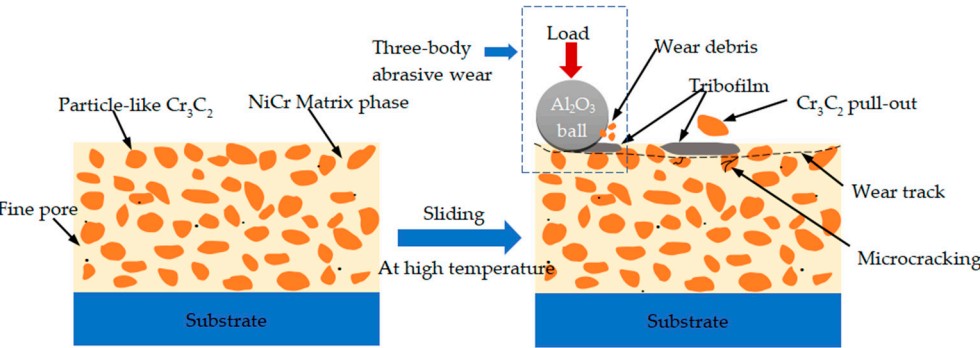

**Figure 13.** Schematic diagram of the wear mechanism of $Cr_3C_2$-25NiCr coating at high temperature.

In the high-magnification SEM images of Figure 12f,h, it can be observed that, at temperatures of 650 °C and 750 °C, some carbide hard particles undergo thermal and shear stresses, gradually developing microcracks under the fatigue action of cyclic load, which eventually causes the hard phase to peel off from the NiCr bonded phase and form pits. Additionally, it can be observed that, during prolonged high-temperature friction and wear processes, a secondary precipitation of the carbides occurs in the NiCr bonding phase [33], potentially leading to the formation of brittle regions and the shedding of carbide hard phases [30].

As the temperature increases from 450 °C to 750 °C, the oxide layer becomes thicker, giving rise to more debris and an oxide-based friction film. Along the edges of the oxide film in Figure 12d,f, there are a large number of fine oxide debris, which will be used as

abrasive particles to form a three-body abrasive wear, resulting in a relatively high wear rate of the coating at 750 °C.

## 4. Conclusions

1.  A $Cr_3C_2$-25NiCr nanoceramic coating was prepared using an air-assisted supersonic flame-spraying technique (HVAF) with a coating thickness of 260 μm. The coating exhibited a dense surface with low porosity (0.34%). The microstructure of the coating displayed a typical thermal spray-layer structure, consisting of a $Cr_3C_2$ hard phase, NiCr bonding phase, and a small amount of $Cr_7C_3$;

2.  The average microhardness of the $Cr_3C_2$-25NiCr coating at room temperature was 998 $HV_{0.3}$, which is approximately five times higher than that of the 316H substrate. The Weibull distribution of hardness values for the coating showed a single peak feature, and small hardness extreme values, indicating stable mechanical performance;

3.  The average microhardness values of the $Cr_3C_2$-25NiCr coating at 450 °C, 550 °C, 650 °C, and 750 °C were 840 $HV_{0.3}$, 811 $HV_{0.3}$, 729 $HV_{0.3}$, and 696 $HV_{0.3}$, respectively. With increasing temperature, the coating exhibited a decreasing trend in microhardness due to high-temperature softening. However, it still maintained relatively high hardness at elevated temperatures;

4.  At 450 °C, the $Cr_3C_2$-25NiCr coating exhibits the best high-temperature friction and wear properties. The wear rates of the coating at 450 °C, 550 °C, 650 °C, and 750 °C were $(2.16 \pm 0.03) \times 10^{-5}$ mm$^3$/(N·m), $(2.52 \pm 0.01) \times 10^{-5}$ mm$^3$/(N·m), $(2.68 \pm 0.01) \times 10^{-5}$ mm$^3$/(N·m), and $(2.97 \pm 0.02) \times 10^{-5}$ mm$^3$/(N·m), respectively. With increasing temperature, the average friction coefficient of the $Cr_3C_2$-25NiCr coating shows a trend of initially decreasing and then increasing, corresponding to the gradual enlargement of the wear scar area;

5.  During high-temperature friction and wear, a large area of sheet-like dark gray oxide film formed on the coating surface. As the temperature increased, the rate of oxide film formation accelerated, and the run-in period of the coating gradually shortened. This oxide film acted as a lubricant between the friction pairs, effectively reducing the friction coefficient of the coating at high temperatures;

6.  The wear mechanism of the $Cr_3C_2$-25NiCr coating at high temperatures is mainly oxidative wear. At temperatures of 650 °C and 750 °C, certain carbide hard particles develop microcracks and undergo secondary precipitation, leading to the formation of a brittle zone. This zone, in conjunction with oxide debris, contributes to the occurrence of three-body abrasive wear.

**Author Contributions:** Writing—original draft, writing—review and editing, Z.Z.; Formal analysis, W.H.; Investigation, S.L.; Data curation, J.Y.; Writing—review and editing, D.S.; Visualization, Y.J.; software, J.X.; Project administration, funding acquisition, Supervision, D.D. All authors have read and agreed to the published version of the manuscript.

**Funding:** This research was funded by Major National Science and Technology Projects, grant number 2019ZX06002026-005, China. And the APC was funded by Chongqing Chuanyi Control Valve Co., Ltd., Chongqing, China.

**Institutional Review Board Statement:** Not applicable.

**Informed Consent Statement:** Not applicable.

**Data Availability Statement:** Not applicable.

**Conflicts of Interest:** The authors declare no conflict of interest.

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
