# Peer review of "Microstructure and High-Temperature Properties of Cr3C2-25NiCr Nanoceramic Coatings Prepared by HVAF"

_coatings, doi:10.3390/coatings13101741_

Round 1
Reviewer 1 Report
- the last four sentences of the abstract should be shortened and written more concise. There are many "and", what additionally makes it a little hard to read
- 2. Materials and Methods, 2.1 Coating preparation: The cleaning process has to be described in detail in order to ensure reproducibility of the results
- line 102 (page 3) Virginia, United States or line 114 Oregon, United States- this cannot be the manufacturer? Who is the manufacturer? This does not become really clear and the mentioned States are rather confusing. This also accounts for any other mentioned instrument. What needs to be mentioned here is the manufacturer name, the instrument name and then the country.
- how was the chemical composition of the 316H stainless steel determined and what is the error of the provided values?
- if Fig.1 b was taken in BSE mode of SEM, should be mentioned in the figures subscription
- Fig. 3: Background should not be subtracted from the diffraction patterns since it gives valuable information on the presence of amorphous material inside the sample. Especially in the case of the investigated samples, the amorphous content might change due to the treatment and if or if not the case, it should be visible from the diffraction patterns.
Fig.13 The figure is really increasing the understanding of the processes, however the wear mechanisms causing a change in the microstructure should be visualized more clearly
Author Response
Point 1: The last four sentences of the abstract should be shortened and written more concise. There are many "and", what additionally makes it a little hard to read.
Response 1: Thank you very much for your advice. I modified these sentences. See lines 22-29 of the article for details.
Point 2: Materials and Methods, 2.1 Coating preparation: The cleaning process has to be described in detail in order to ensure reproducibility of the results.
Response 2: The cleaning process has been added to lines 106-109 of the article.
Point 3: line 102 (page 3) Virginia, United States or line 114 Oregon, United States- this cannot be the manufacturer? Who is the manufacturer? This does not become really clear and the mentioned States are rather confusing. This also accounts for any other mentioned instrument. What needs to be mentioned here is the manufacturer name, the instrument name and then the country.
Response 3: I have revised the relevant content of the instrument name, manufacturer and country as required.
Point 4: how was the chemical composition of the 316H stainless steel determined and what is the error of the provided values?
Response 4: The chemical composition of the 316H stainless steel was determined by ICP-OES and CSA. I have added this information to lines 102-105 of the article.
Point 5: if Fig.1 b was taken in BSE mode of SEM, should be mentioned in the figures subscription.
Response 5: Thank you for the nice reminder. I have changed the figures subscription of Figure 1 to “BSE image of Cr3C2-25NiCr powder (a) surface morphology; (b) cross-sectional morphology”.
Point 6: Fig. 3: Background should not be subtracted from the diffraction patterns since it gives valuable information on the presence of amorphous material inside the sample. Especially in the case of the investigated samples, the amorphous content might change due to the treatment and if or if not the case, it should be visible from the diffraction patterns.
Response 6: Thank you for making this important point. The purpose of removing the background is to reduce the influence of background noise on XRD spectra, making the diffraction peaks of the sample clearer and easier to analyze, thus improving the accuracy of the analysis results. In addition, the analysis of broad peaks can also provide information on some amorphous phases, and this method is used in the article.
Point 7: Fig.13 The figure is really increasing the understanding of the processes, however the wear mechanisms causing a change in the microstructure should be visualized more clearly.
Response 7: Thanks for your suggestion. In future experiments, we will pay more attention to the changes in the microstructure and improve the figure.
Reviewer 2 Report
The authors of the article presented the results of interesting experiments and obtained interesting results. These results have been described in sufficient detail. But there are some questions.
1. It follows from the introduction that the composition of the coating is known. It follows from the Introduction and Section 2.1 that the technology for applying this coating to 316H steel by the HVAF method is known. Even the optimal modes of its deposition on this steel are known. Therefore, this HVAF coated steel is used in practice. In this case, their service life under certain conditions must be known, samples of worn coatings must exist, etc. But then it is not clear why the authors conducted this study.
2. The composition of the coating did not vary in the article, technological parameters did not vary. Coatings obtained by other technologies have not been studied. There are no comparisons in the article.
3. No data on adhesion other than indicating that it is high. Compared to what?
4.Lines 144-149. The friction pair cannot consist of one Al2O3 ball. It is necessary to provide data on the second body. In particular, the material, shape, dimensions of the second body. It is not clear what rotates at a speed of 150 rpm, it is necessary to indicate the diameter and hardness of the Al2O3 ball. Was the rotation circular or helical? Why hasn't gas-abrasive wear been tested? When testing for gas abrasive wear, the lowest wear rate does not necessarily correspond to the highest hardness of the coating, because. abrasive particles could get stuck in the soft components of the coating, hardening them.
5. Line 196-197. Micrometallurgical or mechanical connection? These are different mechanisms.
6. Line 216-217. The authors did not investigate the dependence of properties on porosity, therefore it is not correct to write that a porosity of 0.34% indicates excellent physical and chemical properties.
7. Lines 264-269. The authors note "the high value of high-temperature hardness". However, there is no comparison with other coatings in the article. Without this, one cannot speak of high or low high-temperature hardness. High hardness does not necessarily lead to high wear resistance. It depends on the specific friction conditions.
8. Lines 312-314. You can't write about "excellent wear resistance" without comparing it to another coating. "excellent" is not a scientific characteristic, but an advertising one. A scientific characteristic is "more or less than any value."
9. Lines 422-423. The last sentence in conclusion 4 should be removed, because there is no comparison.
10. Lines 429-431. It is necessary to remove the first sentence of conclusion 6, because there is no comparison.
Author Response
Point 1. It follows from the introduction that the composition of the coating is known. It follows from the Introduction and Section 2.1 that the technology for applying this coating to 316H steel by the HVAF method is known. Even the optimal modes of its deposition on this steel are known. Therefore, this HVAF coated steel is used in practice. In this case, their service life under certain conditions must be known, samples of worn coatings must exist, etc. But then it is not clear why the authors conducted this study.
Response 1: Thanks for your suggestion. We have added the detailed reason for the study in lines 79-87 of the article.
Point 2. The composition of the coating did not vary in the article, technological parameters did not vary. Coatings obtained by other technologies have not been studied. There are no comparisons in the article.
Response 2: This article aims to discuss the microstructure of Cr3C2-25NiCr nano-ceramic coating prepared on the surface of 316H using HVAF technology, as well as its wear resistance at 450 ℃, 550 ℃, 650 ℃, and 750 ℃. The technological parameters used in the experiment have been optimized, and our objective is not to study the influence of coating composition, technical parameters, and different surface treatment technologies on the coating performance. Therefore, no control group has been set up for the aforementioned factors.
Point 3: No data on adhesion other than indicating that it is high. Compared to what?.
Response 3: Thank you for your kind reminders. We have made revisions accordingly.
Point 4: .Lines 144-149. The friction pair cannot consist of one Al2O3 ball. It is necessary to provide data on the second body. In particular, the material, shape, dimensions of the second body. It is not clear what rotates at a speed of 150 rpm, it is necessary to indicate the diameter and hardness of the Al2O3 ball. Was the rotation circular or helical? Why hasn't gas-abrasive wear been tested? When testing for gas abrasive wear, the lowest wear rate does not necessarily correspond to the highest hardness of the coating, because. abrasive particles could get stuck in the soft components of the coating, hardening them.
Response 4: It's our description error, the friction pair has only Al2O3 ball, and 150 r/min is the rotational speed of the disc. The high-temperature friction and wear test in this paper is carried out in high-temperature air, which is very close to the actual working conditions. And we also specified the test medium in lines 160-163.
Point 5: Line 196-197. Micrometallurgical or mechanical connection? These are different mechanisms.
Response 5: Sorry for our inattention. It should be mechanical connection, which I have corrected. See line in 213 of the article for details.
Point 6: Line 216-217. The authors did not investigate the dependence of properties on porosity, therefore it is not correct to write that a porosity of 0.34% indicates excellent physical and chemical properties
Response 6: Thank you for your kind reminders. I revised the wording and introduced references to support my point.
Point 7: Lines 264-269. The authors note "the high value of high-temperature hardness". However, there is no comparison with other coatings in the article. Without this, one cannot speak of high or low high-temperature hardness. High hardness does not necessarily lead to high wear resistance. It depends on the specific friction conditions.
Response 7: Thanks for your suggestion. To make the article more rigorous, I removed ambiguous sentences.
Point 8: Lines 312-314. You can't write about "excellent wear resistance" without comparing it to another coating. "excellent" is not a scientific characteristic, but an advertising one. A scientific characteristic is "more or less than any value."
Response 8: Thank you very much for the reminder. We have made revisions accordingly, see lines 328-329 of the article.
Point 9: Lines 422-423. The last sentence in conclusion 4 should be removed, because there is no comparison.
Response 9: I have revised conclusion 4 and the text to this effect, as detailed in lines 328-329 and 433-434 of the article.
Point 10: Lines 429-431. It is necessary to remove the first sentence of conclusion 6, because there is no comparison.
Response 10: Thank you for your kind reminders. I have deleted the first sentence of conclusion.
Reviewer 3 Report
The paper is logically organized, the results are clearly presented, but the problem of why the authors undertook this research is not defined. The paper acts as a simple case study, the authors prepared the coating, tested it in different conditions and presented the results. What is missing here is the reason, the practical problem they solved, for what application the coating was designed, under what conditions it will work, why they chose an Al2O3 ball as a counterpart in the wear test. Simply, the overall idea of the research is missing. This comment must be taken into account by the authors in the corrected version and the overall research intent must be completed!
For the HVZHT-30 high temperature micro hardness tester, the manufacturer and country are missing.
Please, remove word "This" from title of subchapter 3.1.
Please, how many cross-sections was made for pores observation, value 0.34% of porosity is an average from how many observations?
From line 369 to 389, the authors incorrectly refer to Figure 13d, h, f...
Author Response
Point 1: The paper is logically organized, the results are clearly presented, but the problem of why the authors undertook this research is not defined. The paper acts as a simple case study, the authors prepared the coating, tested it in different conditions and presented the results. What is missing here is the reason, the practical problem they solved, for what application the coating was designed, under what conditions it will work, why they chose an Al2O3 ball as a counterpart in the wear test. Simply, the overall idea of the research is missing. This comment must be taken into account by the authors in the corrected version and the overall research intent must be completed!
Response 1: Thank you for your suggestion. The reason we chose Al2O3 ball is because of its high hardness and ability to achieve better grinding results. And we have added overall research intent in lines 79-87 of the article.
Point 2: For the HVZHT-30 high temperature micro hardness tester, the manufacturer and country are missing.
Response 2: Sorry for our inattention. We have made changes in the article, see the red section in line 154.
Point 3: Please, remove word "This" from title of subchapter 3.1.
Response 3:Thank you for your suggestion. “This” from title of subchapter 3.1 has been removed.
Point 4: Please, how many cross-sections was made for pores observation, value 0.34% of porosity is an average from how many observations?
Response 4:We selected 5 regions on the coating for measurement, resulting in a porosity of 0.34% for the coating. The specific method for measuring the porosity is described in line 133 of the article.
Point 5: From line 369 to 389, the authors incorrectly refer to Figure 13d, h, f...
Response 5: Sorry for our inattention. We have corrected it and it should be Figure 12 (f), (h).
Reviewer 4 Report
This manuscript focuses on the wear, hardness and structural characterisation of chromium carbide NiCr HVAF sprayed coatings. It is relatively well written (I do recommend a finer proofread because there are some syntax errors) and the authors employed a comprehensive characterisation approach. While I believe this manuscript will interest many of Coatings’ readers, I have some comments and questions before recommending it for publication.
Line 183: Have you considered the possibility of nanocrystalline grains to explain peak broadening? Also, the authors claim that the phases present in the powder form are the same present in the coating. However, the XRD shows peaks attributable to the coating that are not present in the powder (e.g. right before and after 30º).
Line 196: The authors claim that the coating is 260 micrometres thick, but the scale in Figure 4 suggests otherwise.
Figure 6: Negative values of hardness have no meaning. I suggest deleting the negative half-axis for hardness from the chart.
Line 267: The authors claim a maintenance of high hardness at elevated temperature. What was the dwell time at high temperature before the hardness measurement? Under service conditions in many applications, substrates and coatings are exposed to high temperatures for long periods of time, hours, days and even weeks! How can authors be sure that for long periods of exposure to high temperatures would still render high coating hardness?
Line 302: Why do the coefficient of friction seems to stabilise at 450 and 750 ºC with time, but not at 550 and 650 ºC, where a decrease with sliding time is observed?
Line 332: What carbides? Could authors provide any evidence for that (XRD, EDS, TEM,…)?
Line 369: There’s no Figure 13d, h. Authors mean Figure 12, perhaps?
There are some things to correct (at least) in lines 96, 126, and 228.
Author Response
Point 1: Line 183: Have you considered the possibility of nanocrystalline grains to explain peak broadening? Also, the authors claim that the phases present in the powder form are the same present in the coating. However, the XRD shows peaks attributable to the coating that are not present in the powder (e.g. right before and after 30º).
Response 1: Thank you very much for your advice. According to our literature accumulation ([15],[17]), in this experiment, the reason for peak broadening is more due to the generation of amorphous phase, so we prefer to use the amorphous phase to explain peak broadening. And we do not indicate in the article that the powder and coating have the same phases, but that the main components of the powder and coating are the same, both Cr3C2 and NiCr. This is the exact words in the article: ” The Cr3C2-25NiCr powder and coating is primarily composed of Cr3C2 hard phase and NiCr bonding phase.”
Point 2: Line 196: The authors claim that the coating is 260 micrometres thick, but the scale in Figure 4 suggests otherwise.
Response 2: Sorry for our inattention. The scale bar should be 100 μm and not 1000 μm, which we have corrected.
Point 3: Figure 6: Negative values of hardness have no meaning. I suggest deleting the negative half-axis for hardness from the chart.
Response 3: Thanks for your suggestion. But we believe that the existence of negative half-axis can well illustrate the influence of HVAF on the hardness of the matrix, and can intuitively explain that after HVAF treatment, the hardness of the matrix has been significantly improved.
Point 4: Line 267: The authors claim a maintenance of high hardness at elevated temperature. What was the dwell time at high temperature before the hardness measurement? Under service conditions in many applications, substrates and coatings are exposed to high temperatures for long periods of time, hours, days and even weeks! How can authors be sure that for long periods of exposure to high temperatures would still render high coating hardness?
Response 4: The holding time at high temperature before hardness measurement is 10 min, which we mentioned that in line 155.
In addition, your suggestion is very constructive, and we will extend the dwell time in subsequent experiments to better discuss the high temperature hardness of the coating over a longer period of time.
Point 5: Line 302: Why do the coefficient of friction seems to stabilise at 450 and 750 ºC with time, but not at 550 and 650 ºC, where a decrease with sliding time is observed?
Response 5: Sorry for our inattention. It's our misnomer. We revised the sentence as follows [lin 316-318]:” Moreover, at this temperature, the friction coefficient was the most stable, fluctuating only within a small range. And with the increase in sliding time, it shows upward trend.”
Point 6: Line 332: What carbides? Could authors provide any evidence for that (XRD, EDS, TEM,…)?
Response 6: The carbides is Cr3C2. Its EDS result is shown in Figure 12(h).
Point 7: Line 369: There’s no Figure 13d, h. Authors mean Figure 12, perhaps?
Response 7: Sorry for our inattention. We revised it to “(as seen in Figure 12 (f), (h))”.
Point 8: There are some things to correct (at least) in lines 96, 126, and 228.
Response 8: Thanks for your kind reminders. We revised the sentence as follows:
” The spraying powder used for the experiments is Cr3C2-25NiCr powder produced by Luoyang Jinglu Hard Alloy Tool Co., Ltd” [lin111-112].
“Microhardness of the coatings was measured using the INNOVATEST FALCON 500 (Innovatest, Eindhoven, The Netherlands) Vickers hardness tester with a load of 300 g and a loading time of 15 s [lin139-140].”
“In the case of HVAF technology, the high powder injection velocity leads to the flattening of most unmelted or semi-melted powder particles. These particles are then layered and bonded onto the substrate surface through the application of significant impact forces and plastic deformation pressures [lin243-247].
Furthermore, we went through the entire manuscript to eliminate grammatical mistakes.
Round 2
Reviewer 2 Report
The reviewer thanks the authors for their clear answers to the reviewer's questions and wishes them further success.
Author Response
Once again, thank you for your invaluable review and I am honored to have benefited from your expertise.